·᷿᷿· PLOS | ONE

# Replacing plasma membrane outer leaflet lipids with exogenous lipid without damaging membrane integrity

**Guangtao Li**[1©], **Shinako Kakuda**[1©], **Pavana Suresh**[1©], **Daniel Canals**[2], **Silvia Salamone**[2], **Erwin London**[1]*

**1** Dept. of Biochemistry and Cell Biology, Stony Brook University, Stony Brook, NY, United States of America, **2** Department of Medicine and Stony Brook Cancer Center, Stony Brook University, Stony Brook, NY, United States of America

© These authors contributed equally to this work.

* Erwin.London@stonybrook.edu

**Data Availability Statement:** All relevant data are within the manuscript and its Supporting Information files.

## Abstract

We recently introduced a MαCD-based method to efficiently replace virtually the entire population of plasma membrane outer leaflet phospholipids and sphingolipids of cultured mammalian cells with exogenous lipids (Li et al, (2016) Proc. Natl. Acad. Sci USA 113:14025–14030). Here, we show if the lipid-to- MαCD ratio is too high or low, cells can round up and develop membrane leakiness. We found that this cell damage can be reversed/prevented if cells are allowed to recover from the exchange step by incubation in complete growth medium. After exchange and transfer to complete growth medium cell growth was similar to that of untreated cells. In some cases, cell damage was also prevented by carrying out exchange at close to room temperature (rather than at 37˚C). Exchange with lipids that do (sphingomyelin) or do not (unsaturated phosphatidylcholine) support a high level of membrane order in lipid vesicles had the analogous effect on plasma membrane order, confirming exogenous lipid localization in the plasma membrane. Importantly, changes in lipid composition and plasma membrane properties after exchange and recovery persisted for several hours. Thus, it should be possible to use lipid exchange to investigate the effect of plasma membrane lipid composition upon several aspects of membrane structure and function.

## Introduction

The ability to readily manipulate the lipid composition of living cell membranes can provide a useful tool for studies of membrane lipid function and lipid-protein interaction. It is possible to alter lipid composition in some cases using lipid synthesis inhibitors. However, they are slow acting, not available for all lipids, and not always specific to a single lipid [1, 2]. Metabolic engineering to alter lipid biosynthesis pathways in living cells can be powerful but is laborious [3]. In addition, these methods also cannot readily be used to introduce unnatural lipids into cells, or introduce lipids with specific headgroup and acyl chain combinations.

**Funding:** Research reported in this publication was supported by the National Institutes of Health R35 GM122493 to Erwin London. The funders had no role in study design, data collection and analysis, decision to publish or preparation of the manuscript.

**Competing interests:** The authors have declared that no competing interests exist.

We previously demonstrated that CDs can be used to carry out efficient outer leaflet phospholipid and sphingolipid exchange in artificial lipid vesicles, such that the entire outer leaflet complement of lipids can be replaced with exogenous lipids [4–10]. Studies by others have been carried out using MβCD or a γCD to introduce phospholipids in intact cells [11–13]. Using MαCD we recently developed a method to replace virtually the entire complement of endogenous plasma membrane outer leaflet phospholipids and sphingolipids with exogenous lipids [14]. The MαCD method does not perturb membrane sterol levels because αCDs do not interact significantly with cholesterol [7, 15, 16].

In this study, the MαCD exchange method was extended to a wider range of conditions by identifying methods to minimize cell damage. In particular, by varying the lipid-to- MαCD ratio, varying exchange temperature, and allowing cells to recover from exchange in complete growth medium we found that efficient replacement of outer leaflet phospholipids and sphingolipids by exogenous lipids can be carried out under conditions that avoid cell damage. Under these conditions, altered lipid composition and altered membrane properties could be maintained over hours, which should be sufficient to allow detailed studies of the effect of plasma membrane outer leaflet lipid structure upon cell function.

## Materials and methods

### Materials

Rabbit blood citrate was purchased from Colorado Serum Company (Denver, Colorado). A549 (ATCC[R]CCL-185[TM]) and Chinese hamster ovary (CCL-61 CHO K-1) cells were obtained from the American Type Culture Collection (Manassas, VA). RBL-2H3 cells were kind gifts from Dr. Barbara Baird (Cornell University). Brain sphingomyelin (bSM), egg sphingomyelin (eSM), 1-palmitoyl-2-oleoyl-sn-glycerol-3-phosphocholine (POPC), and 1,2-dioleoyl-sn-glycero-3-phosphocholine (DOPC) were from Avanti Polar Lipids (Alabaster, AL). Radiolabeled ($^3$H choline methyl) bovine SM and radiolabeled (oleoyl-1-$^{14}$C) POPC were purchased from American Radiolabeled Chemicals (St. Louis, MO). Methyl-α-cyclodextrin (MαCD) was purchased from AraChem (Budel, the Netherlands). RPMI medium 1640, DMEM medium, F12 medium, Opti-MEM, fetal bovine serum, Dulbecco's phosphate-buffered saline (DPBS) (200 mg/L KCl, 200 mg/L $KH_2PO_4$, 8 g/L NaCl, and 2.16 g/L $Na_2HPO_4$), HBSS (Hank's balanced salt solution, pH 7.4, Gibco), penicillin and streptomycin, trypsin with EDTA, Vybrant[R] Apoptosis Assay Kit (Alexa Fluor[R] 488 labeled-Annexin-V/propidium iodide (PI) staining kit) (V13241), Vybrant MTT Cell proliferation assay kit (V-13154), 1-(4-trimethylammoniumphenyl)-6-phenyl-1,3,5-hexatriene p-toluenesulfonate (TMA-DPH) and 6-dodecanoyl-2-dimethyaminonaphthalene (Laurdan) were purchased from Thermo Fisher Scientific (Waltham, MA). Bovine serum albumin (BSA) was obtained from Millipore (Kankakee, IL).

### Preparation of lipid-loaded MαCD

Preparation of lipid-loaded MαCD was described previously[14]. Briefly, the desired amount of lipids were dried and multilamellar vesicles were prepared by adding medium (DPBS for red blood cells, Ham's F-12 with L-glutamine for CHO cells, DMEM for RBL-2H3 cells, or RPMI 1640 for A549 cells) prewarmed to 70°C for 5 min. Then MαCD from a ~300–400 mM stock solution dissolved in DPBS was added to the desired concentration. The mixture of vesicles and MαCD was incubated at 37°C for 30 min. The lipid-loaded MαCD was then used for lipid exchange.

## Lipid exchange in red blood cells

Rabbit red blood cells (RBC) were prepared from whole blood by centrifuging blood at 1000 x g for 3 min. After centrifugation, the supernatant was removed, RBCs were then washed three times with the initial blood volume of DPBS, and then resuspended to the initial blood volume using DPBS. To carry out exchange, a 10 μl aliquot of the RBCs were added to 990 μl of the lipid-loaded MαCD and incubated for 1 h in a 37°C incubator. After exchange, cells were centrifuged at 1000 x g for 2 min. A 50 μl aliquot of the supernatant was diluted with 950 μl DPBS and used to determine absorbance of hemoglobin at 540 nm using Beckman DU650 spectrophotometer.

## Cell culture

A549 cells were cultured in RPMI 1640 supplemented with 10% (by volume) fetal bovine serum (FBS). RBL-2H3 cells were cultured in DMEM medium supplemented with 10% FBS. CHO cells were grown in Ham's F-12 supplemented with 10% FBS. In all cases 100 U/ml penicillin and streptomycin were also present. All cells were maintained at 37°C in an incubator with 5% $CO_2$.

## Lipid exchange in cultured cells and cell recovery

Cells grown to 80–90% confluency were washed in DPBS three times and lipid-loaded MαCD was added (2 or 3 ml/10 cm diameter plates, 1.5 ml/6 cm diameter plates, or 1 ml/3.5 cm diameter plates, as indicated in figures). The MαCD concentration used was 40 mM for A549 cells and 50 mM for CHO cells, unless otherwise noted. Exchange carried out for 1 h at the desired temperature (37°C unless otherwise noted). After exchange, the lipid- MαCD mixture was removed, and cells were washed in DPBS three times. To allow cells to recover from exchange, complete (serum-containing) growth medium was added to the cells (2 ml or 10 ml/10cm diameter plates, 2ml/6cm diameter plates or 1.5ml/3.5 cm diameter plates, as indicated in figures). Samples were then incubated at 37°C with 5% $CO_2$ for 2 h, unless otherwise noted. After this incubation, cells were washed with DPBS before further analysis.

## TLC analysis of lipids

TLC was carried out similarly as previously [14]. Briefly, lipids were extracted from cells after removal of medium and washing with DPBS using 3:2 (v:v) hexanes:isopropanol. The extracted lipids were then dried under $N_2$ and dissolved in 1:1 v/v chloroform:methanol before an aliquot containing 10–20% of total lipid was chromatographed on silica gel TLC plates (HP-TLC Silica Gel 60 plates (Merck)) in 65:25:5 (v:v) chloroform/methanol/28.0–30.0 (v/v)% ammonium hydroxide. For measuring radioactivity, the plates were stained with iodine and the lipid bands removed and added to scintillation fluid. For measuring lipid in non-radioactive samples, the lipids were charred and band intensity measured as described previously [14].

## Annexin-V and propidium iodide staining

Alexa Fluor 488 labeled-Annexin-V and propidium iodide (PI) staining kits were utilized as biomarkers for the detection of phosphatidylserine exposure and cell membrane integrity, according to the manufacturer's protocols (Invitrogen, Carlsbad, CA). Briefly, after lipid exchange, cells were trypsinized, washed with DPBS and then resuspended in 100μl of binding buffer (50 mM HEPES, 700 mM NaCl, 12.5 mM CaCl₂, pH7.4) per million cells. After addition of Annexin-V and PI, cells were incubated at room temperature for 15 min while protected

from light. Samples were diluted with 0.4ml of binding buffer per million cells, and analyzed using a FACSCalibur (BD, Bioscience) flow cytometer and CellQuest Pro software (BD, Bioscience) or FlowJo version 10 software, collecting 10,000 or 50,000 events per sample. PI fluorescence was measured with a 488 nm laser excitation and a 650LP filter. Annexin V fluorescence was measured exciting with a 488 laser and measuring emission with a 530 filter. The % damaged cells equals the amount of double PI and annexin V cells plus the amount of singly positive cells divided by total cells. (The number of singly annexin V positive cells was so low, they were not included in most of the calculations.)

### Assessment of cell viability with trypan blue

To assess cell membrane intactness based on Trypan Blue exclusion, the cells were detached with trypsin-EDTA (Gibco) and centrifuged at 500 x g for 5 min. 200μl of a 0.2 (w/v) % Trypan Blue in DPBS was then added, and cells that excluded Trypan Blue were counted using a Countess automated cell counter (Invitrogen). The ratio of cells excluding Trypan Blue in lipid-exchanged samples to cells excluding Trypan Blue in untreated samples was calculated.

### Measurement of cell proliferation

Vybrant MTT Cell proliferation assay kit (V-13154) (Invitrogen) was used for determining cell growth rates. CHO cells or A549 cells before or after lipid exchange were seeded in 96-well plates, at a density of 8,000 cells/well, and 0.20 ml of complete growth medium added. The assay was carried out after the desired period of cell growth according to manufacturer's instructions.

### Measurement of GFP leakage

CHO cells cultured in a 3.5-cm plate were transfected with 1μg of plasmid encoding GFP (pEGFP-N1) (Clontech) using 6 μl of polyethylenimine (PEI) reagent mixed with 100 μl Opti-MEM (Gibco). After 28–30 h post-transfection, exchange with 3.0 mM eSM/50 mM MαCD was performed. For recovery, after washing three times with DPBS, complete growth medium was added and cells were cultured at 37˚C for 2 h. After removing the medium and washing with DPBS cells were detached with 2ml of Enzyme-free cell dissociation solution (Millipore, Kankakee, IL) and resuspended in 0.5 ml Hank's balanced salt solution (HBSS) pH 7.4 (Gibco) containing 1% (w/v) BSA. Next, 1 μl of 0.1 mg/ml PI/water was added to 500 μl of cell suspension. GFP fluorescence was measured exciting with a 488 laser and emission measured with a 530 filter. For each sample,10,000 events were collected and both the percentage of PI and GFP positive cells on a FACSCalibur (BD, Bioscience) flow cytometer using CellQuest Pro software (BD, Bioscience).

### Characterization of plasma membrane properties

With or without 1 h lipid exchange with the desired lipids and 2 h recovery in complete growth medium, A549 cells were trypsinized and resuspended in DPBS at a density of 0.5 million cells per ml. To 1 ml samples 10 μl aliquots of TMA-DPH or Laurdan, dissolved in ethanol, were added to a final concentration of 40nM or 400nM, respectively. After a 10 min incubation at room temperature, Laurdan emission spectra were measured from 400 nm to 550 nm with an excitation wavelength of 385 nm. GP values were calculated as described previously [17]. Fluorescence anisotropy was measured for samples with TMA-DPH as described previously [9].

### Characterization of GPMV properties

Giant plasma membrane vesicles (GPMV) were prepared as in published protocols [17]. Briefly, RBL-2H3 cells, with and without 1h lipid exchange and 2h recovery in complete

growth medium, were washed three times with DPBS and once with GPMV buffer (10 mM HEPES, 150 mM NaCl, 2 mM CaCl$_2$, pH 7.4). Then 2 mL of freshly prepared GPMV buffer with 2 mM DTT and 25 mM paraformaldehyde (PFA) was added, and samples incubated at 37˚C for 1 h, rocking every 5 min. The GPMV-containing solution was removed from the cells, and then centrifuged at 100 x g for 10 min to pellet cell debris. GPMV concentration was estimated from the fluorescence intensity of DPH added to the GPMV in excess relative to DPH fluorescence in a standard curve of various concentrations of 1:1 mol:mol SM:POPC vesicles [18]. GPMV experiments were carried out at a final lipid concentration of 4–8 μM. TMA-DPH or Laurdan were added to the GPMV and Laurdan emission spectrum and TMA-DPH anisotropy were measured as described above.

## Analysis of sphingomyelin profile in A549 cells using mass spectrometry

After lipid exchange, A549 cells were washed four times with DPBS. Cells were either immediately subjected to lipid extraction or subjected for lipid extraction after 1, 2 or 4 h recovery in complete growth medium at 37˚C. Control A549 cells not subjected to lipid exchange were also prepared. Cells were washed four times with cold PBS and directly collected from the dish using solvent A (1.4:0.6:3 v/v isopropanol:water:ethyl acetate). A mix of C17-sphingolipids were used as internal standard with 50 pmol of each added to each sample. Samples were centrifuged at 2000 x g, and the upper phase transferred to a glass tube. The lower phase was re-extracted adding 2 ml of solvent A, combined with the first upper phase, and dried under N$_2$. Samples were resuspended in 150 μl of methanol. Identification of sphingolipid species was performed on an Agilent Quadropole 6020 LC/MS system and confirmed on a Thermo TSQ Quantum Ultra triple quadrupole MS operating in a multiple reaction monitoring positive ionization mode as described previously [19]. Retention times and calibration curves were determined for each analyzed sphingolipid species using standards from Avanti Polar Lipids (Alabaster, AL). Results from MS analysis were normalized by protein content calculated using the BCA assay (Thermo Fisher Scientific) following the manufacturer's protocol.

## Analysis of exogenous SM exchangability after delivery into A549 cells

40 mM MαCD was pre-incubated with 3 mM 1:1 (mol:mol) bSM:POPC plus 0.5 x 10$^6$ cpm $^3$H-SM or with 3 mM POPC plus 0.5 x 10$^6$ cpm $^{14}$C-POPC. A549 cells cultured in 35 mm plates were incubated with these mixtures for 1 h either at 37˚C (bSM:POPC mixture) or 25˚C (POPC). After washing 4 times with DPBS, the cells were subjected to a second round of lipid exchange either right after the initial exchange or after an additional 2 h recovery incubation in RPMI 1640 medium with 10% FBS added. The second lipid exchange was carried out with 40 mM MαCD pre-incubated with 3 mM 1:1 (mol:mol) bSM:POPC either at 37˚C (for bSM: POPC mixture) or 25˚C (for POPC). For background samples in the first round of exchange, the cells were treated with 3 mM 1:1 (mol:mol) bSM:POPC and 0.5x10$^6$ cpm $^3$H-SM at 37˚C or with 3 mM POPC and 0.5 x 10$^6$ cpm $^{14}$C-POPC without MαCD for 1h at 25˚C. In another experiment cells were treated with 3 mM 1:1 (mol:mol) bSM:POPC without MαCD in the second round of exchange. The lipids were extracted from cells in 1 mL 3:2 (v:v) hexanes:isopropanol and dried under N$_2$. Then lipids were dissolved in 100 μL 1:1 (v:v) chloroform: methanol and radioactivity measured by scintillation counting. % radiolabeled lipid removed after second exchange = (CPM after first round of exchange—CPM after second round of exchange)/(CPM after first round—CPM background) x 100%. This calculation gives total back exchangability. It should be noted that there was a small decrease of CPM in the absence of MαCD, which may indicate there is a small of amount back exchange even in the absence of MαCD.

## Results

### Effect of MαCD and lipid on RBC demonstrates lipid balance upon exchange is important to prevent cell damage

As noted above, our prior study of lipid exchange using MαCD, which was able to efficiently interact with and exchange a wide variety of phospholipids, found that incubating mammalian cells with a mixture of MαCD and exogenous phospholipid/sphingolipid induced efficient lipid exchange, in which over a wide range of conditions virtually the entire lipid monolayer of the outer leaflet was replaced by exogenous lipid [14]. It was briefly noted that at certain lipid and MαCD concentrations cells rounded up and detached from plates [14]. Thus, it is important to understand what lipid and MαCD concentrations avoid perturbing/damaging cells when carrying out lipid exchange. First, we used rabbit red blood cell (RBC) lysis to investigate what lipid and MαCD concentrations damage cells. Lysis was detected by hemoglobin release. In the absence of exogenous lipid, MαCD induced membrane lysis (Fig 1, squares). Lysis by MαCD, which does not bind cholesterol, very likely reflects membrane damage due to extraction of plasma membrane outer leaflet phospholipids (glycerophospholipids and SM). Loss of outer leaflet lipid would lead to an imbalance in the amount of lipid in inner and outer leaflets, which would destablize the plasma membrane. It would be expected that delivery of exogenous lipid to RBC would compensate for the lipid extracted by MαCD. Consistent with this hypothesis, when RBC were incubated in mixtures of exogenous SM and MαCD, the amount of lysis was reduced (Fig 1, triangles).

The concentration of exogenous lipid had a strong effect on the extent of lysis. Using 55 mM MαCD there was minimal lysis with 1.5 mM exogenous bSM (Fig 1), but more lysis at both higher and lower bSM concentrations. Presumably, lipid delivery and extraction are balanced at 1.5 mM bSM. At higher MαCD concentrations lipid extraction from the outer leaflet could exceed lipid delivery, and at lower MαCD concentrations there may be too much delivery of lipid into the outer leaflet. Both of these conditions could lead to lysis.

It should be noted that the amount of imbalance between lipid delivery and extraction leading to lysis may be only a few percent, because lipid bilayers can only stretch a few percent before membrane integrity is lost [20]. A small level of lipid imbalance would be difficult to measure. However, the hypothesis that balance between lipid extraction and delivery is crucial makes two readily-tested predictions. These are that: 1. at high MαCD concentration, decreasing exogenous lipid concentration should increase lysis. 2. at low MαCD concentration, increasing exogenous lipid concentration should increase lysis. This is what is observed. At 70 mM MαCD lysis increases as eSM is decreased from 3 mM to 1.5 mM (Fig 2 inset, solid arrow). At 25 mM MαCD lysis increases as eSM is increased from 0.5 mM to 1.5 mM (Fig 2 inset, dashed arrow). As another way to see this, notice that no/minimal lysis was observed at an intermediate concentration of MαCD, and that when the amount of exogenous SM was increased from 0.5 mM to 3 mM, the concentration of MαCD giving minimal lysis increased from 35 mM to 65 mM (Fig 2, inset). In other words, when exogenous SM concentration is increased the amount of MαCD needed to achieve balance between lipid extraction or delivery increased, as expected.

### Effect of MαCD and lipids upon cultured mammalian cells shows cell damage if there is lipid imbalance upon exchange

The generality of the behavior observed in RBC was investigated in cultured mammalian cells. When exogenous lipid concentration was varied in mixtures with a fixed MαCD concentration, and the MαCD-lipid mixture incubated with Chinese hamster ovary (CHO) cells, cell morphology was altered at both low and high lipid concentrations, with cells becoming rounded, but not at intermediate lipid concentrations (Fig 3A). Rounding up is suggestive of

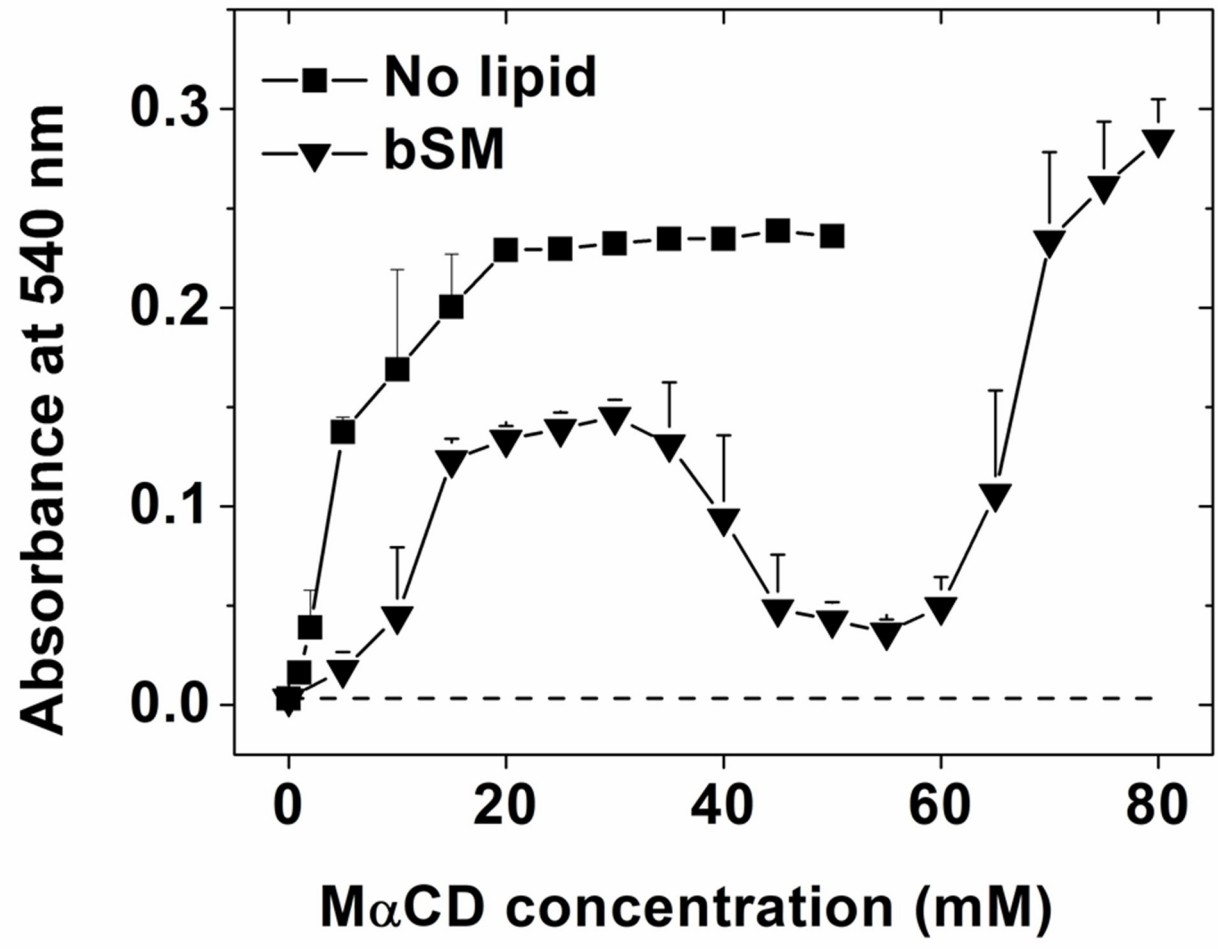

**Fig 1. MαCD concentration and lipid affect rabbit RBC lysis.** Absorbance at 540 nm detects hemoglobin released after cell lysis. Lysis measured in the presence of MαCD (squares) or a mixture of 1.5 mM bSM with MαCD (triangles). The dotted line represents untreated RBC control. Mean and standard deviation from three experiments is shown.

weakened/fewer interactions with the surface of the culture plate, and consistent with this there was detachment of cells from wells at low and high exogenous lipid concentrations (Fig 3B). At extreme lipid concentrations detachment was complete. To see if changes in morphology were associated with damage to membrane integrity as in RBC, leakage of propidium iodide into cells was measured. The binding of annexin V, usually indicative of the appearance of PS in the plasma membrane outer leaflet, i.e. a loss of membrane asymmetry, was also measured. Fig 3C shows examples of raw flow cytometric analysis of cells before and after incubation with MαCD and exogenous SM, and Fig 3D summarizes this by graphing the total fraction of cells with membrane damage (high amounts of PI staining and/or annexin V staining) after incubation with MαCD and exogenous SM. Fig 3D shows that membrane damage was observed at low and high exogenous SM concentrations, but not at intermediate concentrations, analogous to what was observed with RBC. Exchange with brain SM resulted in less membrane damage than exchange with egg SM. Cells with altered shape tend to be the ones that showed cell damage (S1 Fig). In agreement with these results, trypan blue exclusion from cells decreased at low and high exogenous SM concentrations, and trypan blue exclusion was greater with brain SM than with egg SM (S2 Fig).

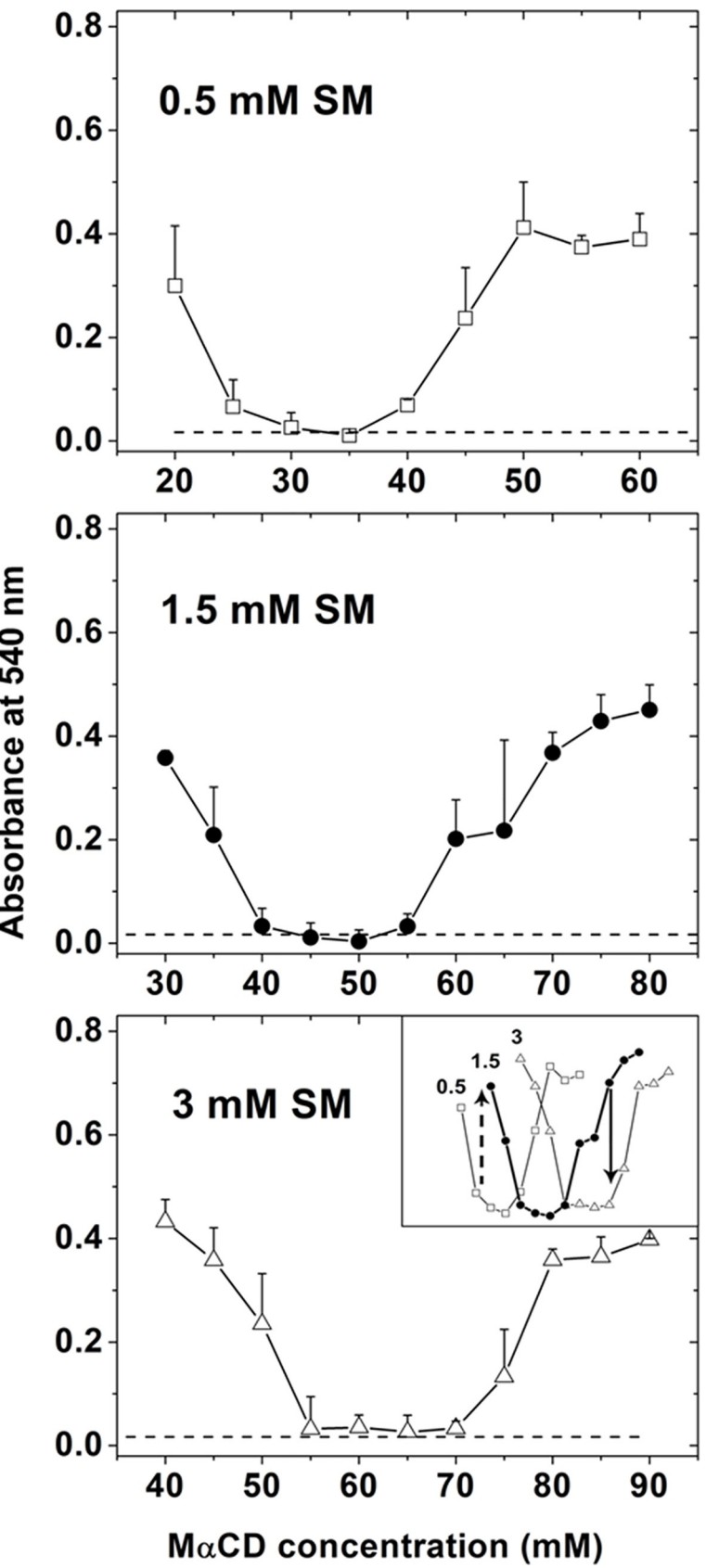

**Fig 2. Effect of lipid exchange on rabbit RBC lysis at varying MαCD and eSM concentrations.** Exchange with eSM is shown at: 0.5 mM eSM (top, squares), 1.5 mM eSM (middle, circles) and 3 mM eSM (bottom, triangles). The inset shows comparison of lysis at these three eSM concentrations. Untreated RBC control is shown as dashed lines. Extent of lysis assayed by release of hemoglobin monitored at 540nm. Mean and standard deviation from three experiments is shown.

## Allowing cells to recover in complete growth medium prevents/reverses cell damage

When CHO and human A549 lung cancer cells were incubated in complete growth medium after lipid exchange, growth curves were similar to those for untreated cells (Fig 4). Thus, it seemed possible that incubation of cells in complete growth medium might prevent or minimize membrane damage. To test this, membrane damage was measured after adding a "recovery" incubation step in complete growth medium after the lipid exchange step. Fig 5A shows that upon incubation in complete growth medium subsequent to exchange under membrane damaging conditions, CHO cells resumed their normal morphology, and Fig 5B and 5C show that after incubation in complete growth medium the fraction of cells damaged upon lipid exchange decreased to near-baseline levels.

The recovery incubation minimized CHO cell membrane damage after exchange over a wide range of exogenous SM concentrations (Fig 6A). A549 cells showed behavior upon eSM exchange similar to that of CHO cells both in terms of membrane damage, and its being largely eliminated by including a recovery step in complete growth medium (Fig 6B). Interestingly, A549 cells did not exhibit membrane damage after bSM exchange over the concentration range tested.

It should be noted that observation of membrane damage as detected by PI leakage into cells after exchange does not necessarily mean that the cells were leaky after exchange. Instead, membrane integrity could be weakened after exchange, with the processing steps needed to assay PI leak into cells (i.e. cell washing with DPBS) being sufficient to destroy membrane integrity. This was tested by experiments which leakage of GFP (MW 37K) *out* of cells was measured. After exchange and washing carried out on GFP-transfected CHO cells using sub-optimal conditions, CHO cells lost most of their endogenous GFP (Fig 7). This shows membrane damage was sufficient to let molecules as large as GFP leak out. However, a recovery incubation reduced the amount of GFP that leaked out of cells. If the GFP leakage occurred during the exchange step itself, the amount of GFP lost after the recovery step should at least equal that lost after exchange. Since this is not observed, it suggests washing the cells was responsible for most of the loss of GFP after exchange. However, the observation of some GFP leakage after a recovery step must involve a loss of GFP at some earlier step in the exchange protocol, since PI, which is smaller than GFP does not leak in after recovery. A guess is that the loss of GFP may reflect some membrane disruption occurring during replacement of the exchange solution with complete growth medium. In any case, it is clear that membrane integrity is weakened upon lipid exchange under sub-optimal conditions, and this is reversed after the recovery incubation. It should be noted that observation of GFP leakage complicates interpretation of annexin V binding results. Annexin V binding could reflect the appearance of PS on the outer surface of the plasma membrane, or that annexin V (MW 36kD) is leaking into the cells and binding PS on the cytosolic side of the membrane.

## Use of lower temperatures allows exchange with unsaturated PC avoiding membrane damage

The studies above were carried out using exogenous SM. We previously showed that exogenous SM/PC mixtures can also be used [14]. To change membrane composition and properties

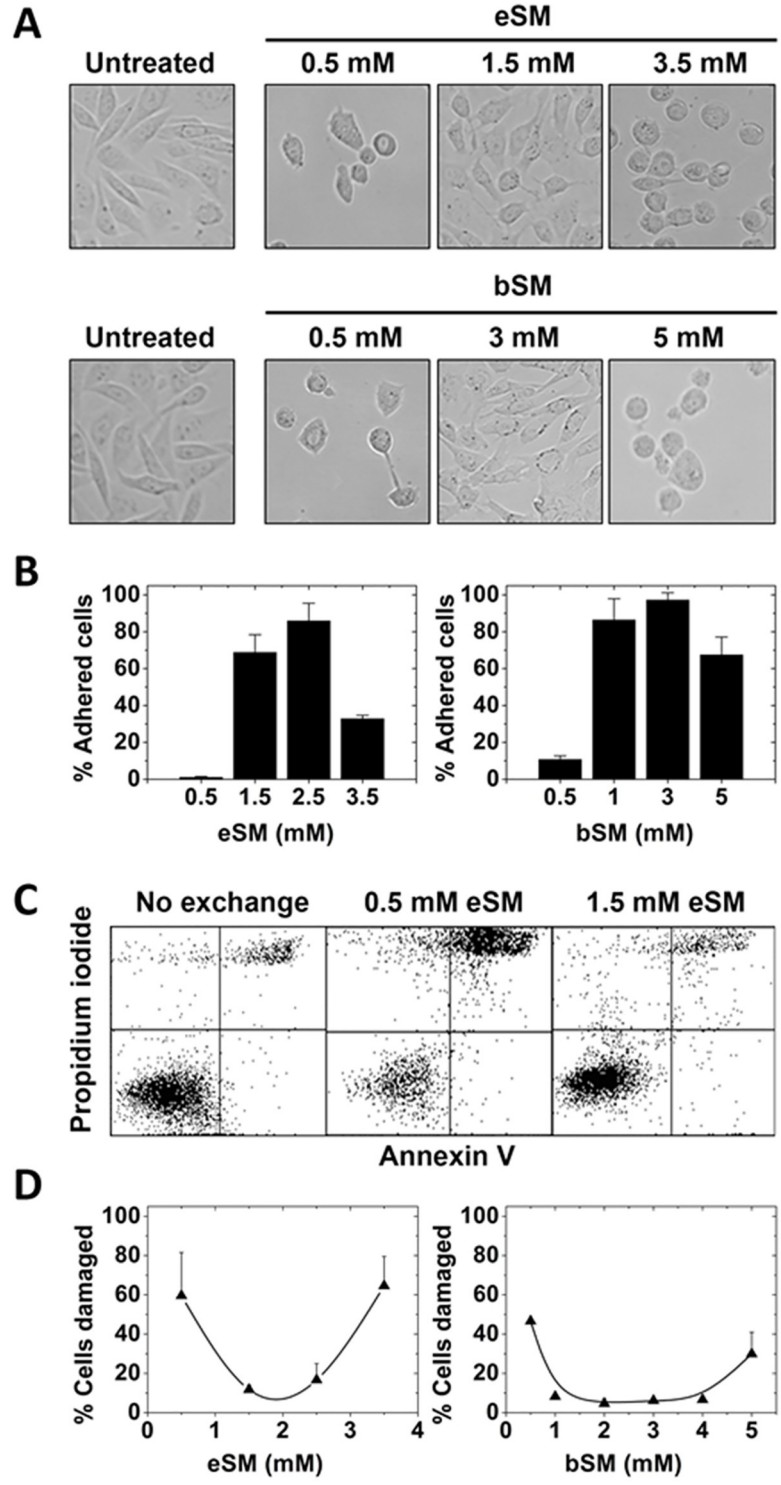

**Fig 3. SM concentration used for lipid exchange affects cell morphology and membrane damage of CHO cells.** A. Cell morphology. Bright-field microscopy images of cell morphology after exchange at indicated concentrations. B. Fraction of cells attached to plates after lipid exchange and two washes with DPBS. (See S5 Fig. for raw data with axis values.) C. Example of dot-plot graph of flow cytometric analysis of PI and Annexin V staining after lipid exchange. D. % of cells damaged after exchange as judged by PI and annexin V staining (see Methods). Exchange was carried out in 3.5 cm diameter plates at 37°C with 1 ml of lipid plus 50 mM MαCD. For D. mean and standard deviation from three experiments is shown.

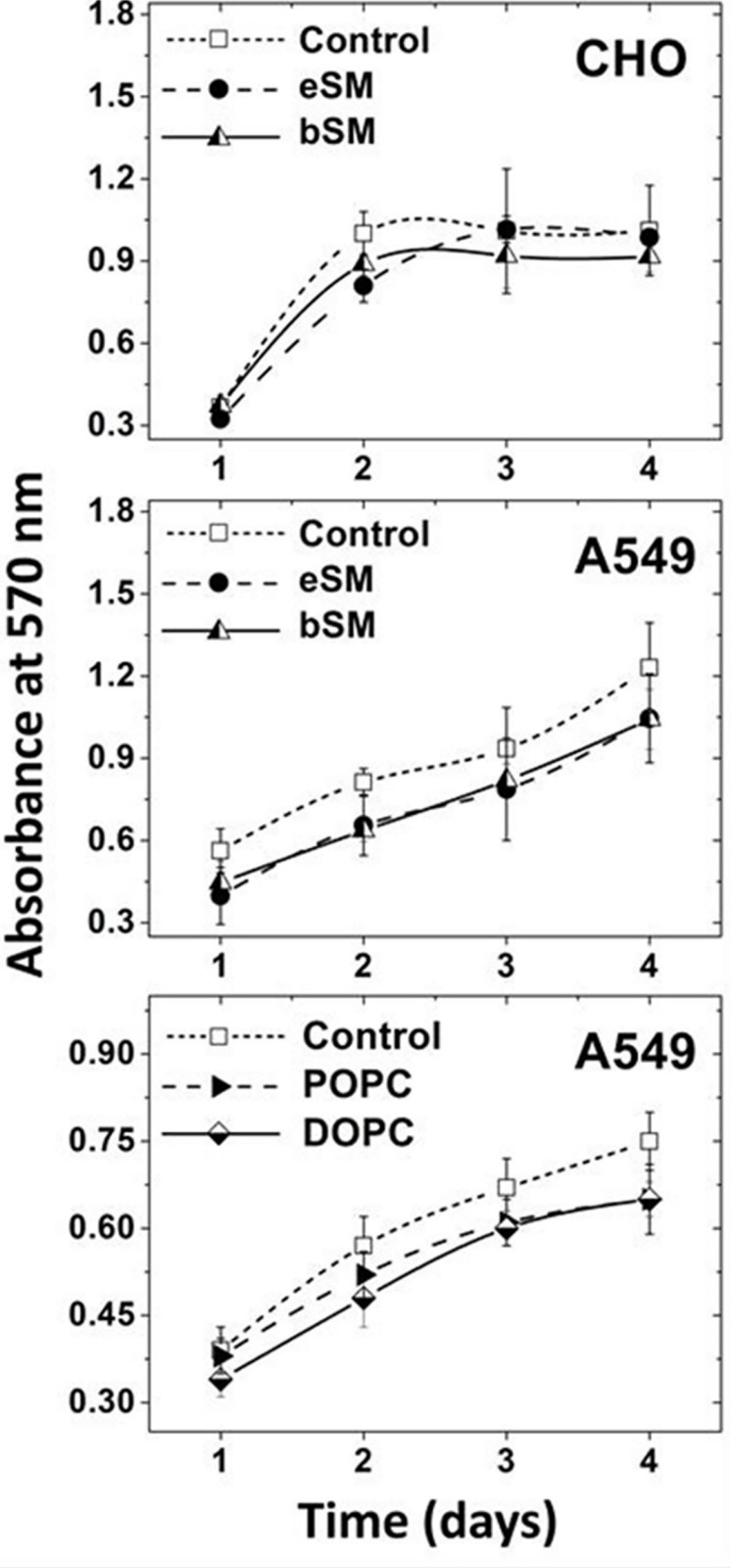

**Fig 4. Proliferation of CHO and A549 cells is not affected by lipid exchange.** Time-course is shown for change of cell number for untreated cells, and cells after exchange, as measured by the MTT assay. Exchange was carried out in

3.5 cm diameter plates at 37˚C with 1.5 mM SM or 4 mM PC plus 40 mM MαCD (for A549 cells) or with CHO cells using 1.5 mM SM and 50mM MαCD. Cells were grown in 96 well plates with 0.2 ml of complete growth medium (see Methods). Mean and standard deviation from three experiments is shown.

over a wider range, it was desirable to extend the method to the use of unsaturated PC species without SM. However, preliminary studies indicated that this was difficult to do while avoiding membrane damage. We found this could be circumvented by carrying out lipid exchange at close to room temperature (24–27˚C) instead of at 37˚C. Fig 8 shows lipid exchange at 26˚C using exogenous POPC or DOPC resulted in only a low amount of cell damage. Adding a recovery step did not change the level of cell damage. Fig 8 also shows exchange at 26˚C with a

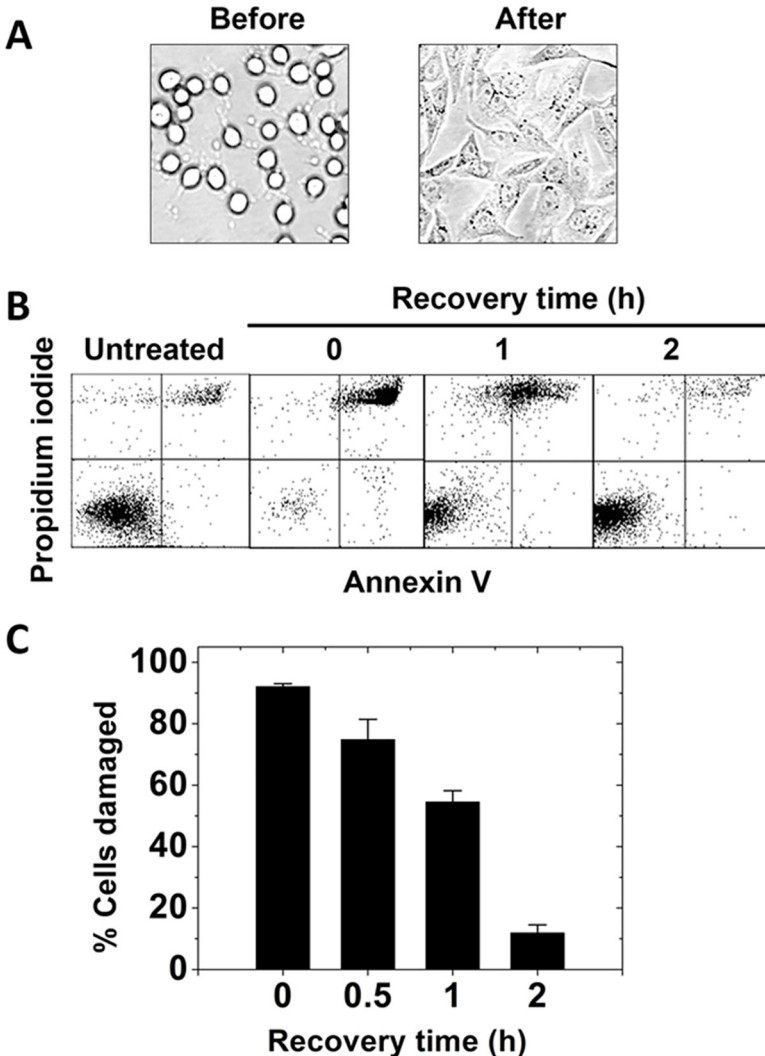

**Fig 5. Recovery in complete growth medium reverses/prevents cell damage.** A. Bright-field image after 5 mM bSM exchange (before) and after 2h recovery in complete growth medium (after) for CHO cells. B. Examples of dot-plot of flow cytometric analysis of PI and Annexin V staining of CHO cells after exchange with bSM and incubation at different recovery times. (See S5 Fig for raw data with axis values.) C. Fraction of damaged cells (PI positive + PI and annexin V double positive cell) from flow cytometric analysis. Exchange was carried out in 3.5 cm diameter plates at 37˚C with 1 ml of lipid plus 50 mM MαCD. Recovery was carried out at 37˚C in the same plates with 1.5 ml of complete growth medium. Mean and standard deviation from three experiments is shown.

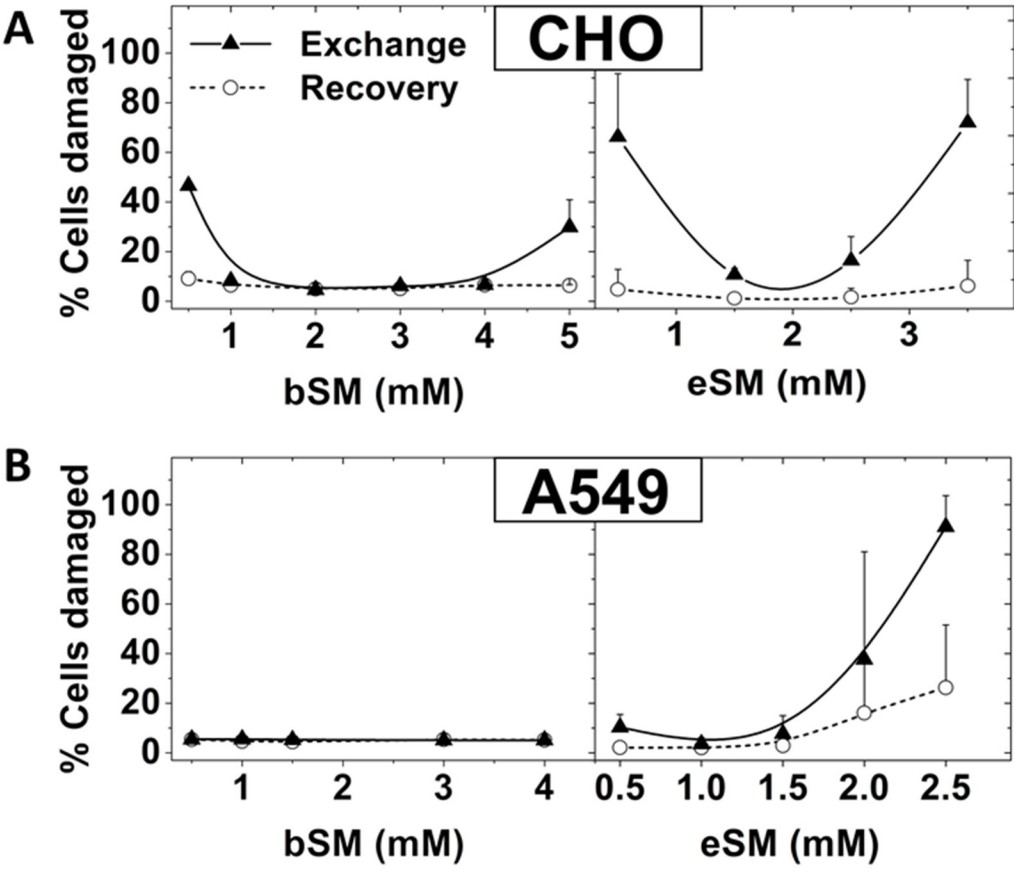

**Fig 6. Effect of recovery step upon cell damage after exchange at different lipid concentrations.** Lipid exchange carried out at 37˚C for 1 h on A. CHO or B. A549 cells using MαCD loaded with different concentrations of eSM or bSM. Cells were subjected to PI and Annexin V staining immediately after lipid exchange (triangles) or after a 2h recovery in complete growth medium (open circles). Exchange was carried out in 3.5 cm diameter plates at 37˚C with 1 ml of lipid plus 40 mM MαCD for A549 cells or 50 mM MαCD for CHO cells. Recovery was carried out at 37˚C in the same plates with 1.5 ml of complete growth medium. Mean and standard deviation from three experiments is shown.

low amount of cell damage after a recovery step could be achieved with exogenous SM. The difference between exchange at low and high temperature did not seem to involve a difference in the rate of exchange, as S3 Fig. shows replacement of endogenous SM in A549 cells occurred at a similar rate at 27˚C and 37˚C.

## Incubation with MαCD and lipids induces efficient exchange of plasma membrane lipids that is persistent

That lipid exchange was efficient when cells are incubated with MαCD-lipid mixtures was demonstrated in our previous report [14]. However, because of the modification of the exchange protocols used here, changes in lipid composition after exchange and recovery were studied in CHO cells in more detail. Using radiolabeled cells, endogenous radiolabeled lipid removal by exchange with exogenous eSM was measured at an eSM concentration in which membrane integrity was not compromised by exchange (Fig 9). In agreement with our previous results [14], about 80% of endogenous SM was removed in 60 min, together with removal of a much smaller fraction of PC. We noted previously that 80% is the limit of SM that can be exchanged even when increasing exchange time, exogenous lipid concentration, lipid type and concentration of MαCD, strongly suggesting that under these conditions outer leaflet

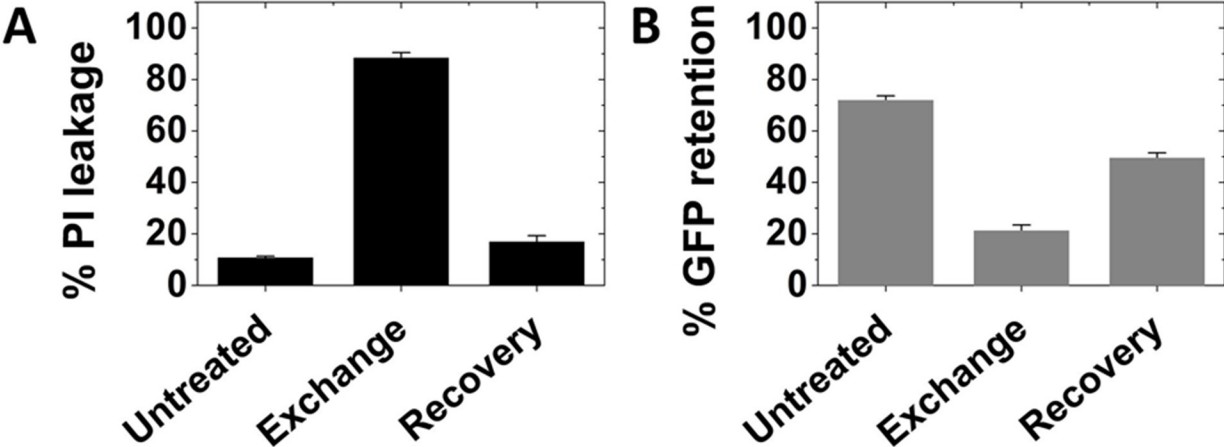

**Fig 7. Recovery minimizes the loss of membrane integrity after lipid exchange.** CHO cells were transfected with eGFP expression plasmid 1d before exchange at conditions sub-optimal for maintaining membrane integrity. A. PI leakage and B. GFP retention analyzed by flow cytometry in untreated cells, immediately after exchange, or after a 2h recovery step. Exchange was carried out in 3.5 cm diameter plates at 37˚C with 1 ml of 3 mM SM plus 50 mM MαCD. Recovery was carried out at 37˚C in the same plates with 1.5 ml of complete growth medium. Mean and standard deviation from three experiments is shown.

exchange is complete. (However, any pool of outer leaflet lipids tightly bound to protein and with a dissociation rate constant of hours or more would not be exchanged.) The lower amount of PC removed is also consistent with previous results, and reflects its localization in many membranes in addition to the plasma membrane outer leaflet. The small decreases in PE and PS+PI, which are not abundant in the plasma membrane outer leaflet likely largely reflect a small amount of cell loss as exchange proceeds (see above).

The efficiency of exchange was further confirmed by considering the amount of lipid exchanged into cells. The amount of SM in A549 cells and delivered to them after exchange and recovery was measured using mass spectrometry. Fig 10 shows the acyl chain structure of the SM delivered into the cells reflects that of the exogenous SM used. In addition, Fig 10 shows that total SM levels increased ~2.2 fold after exchange. This increase is consistent with what we observed previously [14], and is indicative of near total replacement of endogenous outer leaflet phospholipid. (We previously found the plasma membrane outer leaflet of A549 cells was about 40% SM [14]. Thus, SM in the outer leaflet should increase 2.5-fold if exchange with exogenous SM is complete. If the 80% exchangeable endogenous SM represents the outer plasma membrane leaflet pool, total SM should increase 2.2-fold after complete exchange of the outer leaflet lipids with exogenous SM.) Interestingly, the elevated level of SM was persistent, with only a modest decrease in SM levels during the recovery step, and no obvious large remodeling of SM acyl chain structure. The latter is consistent with prior studies showing phospholipids with natural acyl chains are not rapidly remodeled when introduced into cells using MβCD [13].

These results were confirmed for exchange at lower temperature by TLC analysis of lipid composition. The extent of lipid exchange at 27˚C after a 2h recovery was assayed using TLC by the change in the intensity of staining of SM bands (S4 Fig). After exchange with SM loaded MαCD, total cell SM increased by ~2–2.5 fold, similar to what was observed for exchange at 37˚C, while incubation with PC decreased endogenous SM levels relative to no MαCD controls by roughly ~60–70%, slightly less than the loss of radiolabeled endogenous SM when exchange was carried out at 37˚C. Thus, it appears that removal of the outer leaflet lipids and their replacement with exogenous lipids is also complete or near complete when exchange is

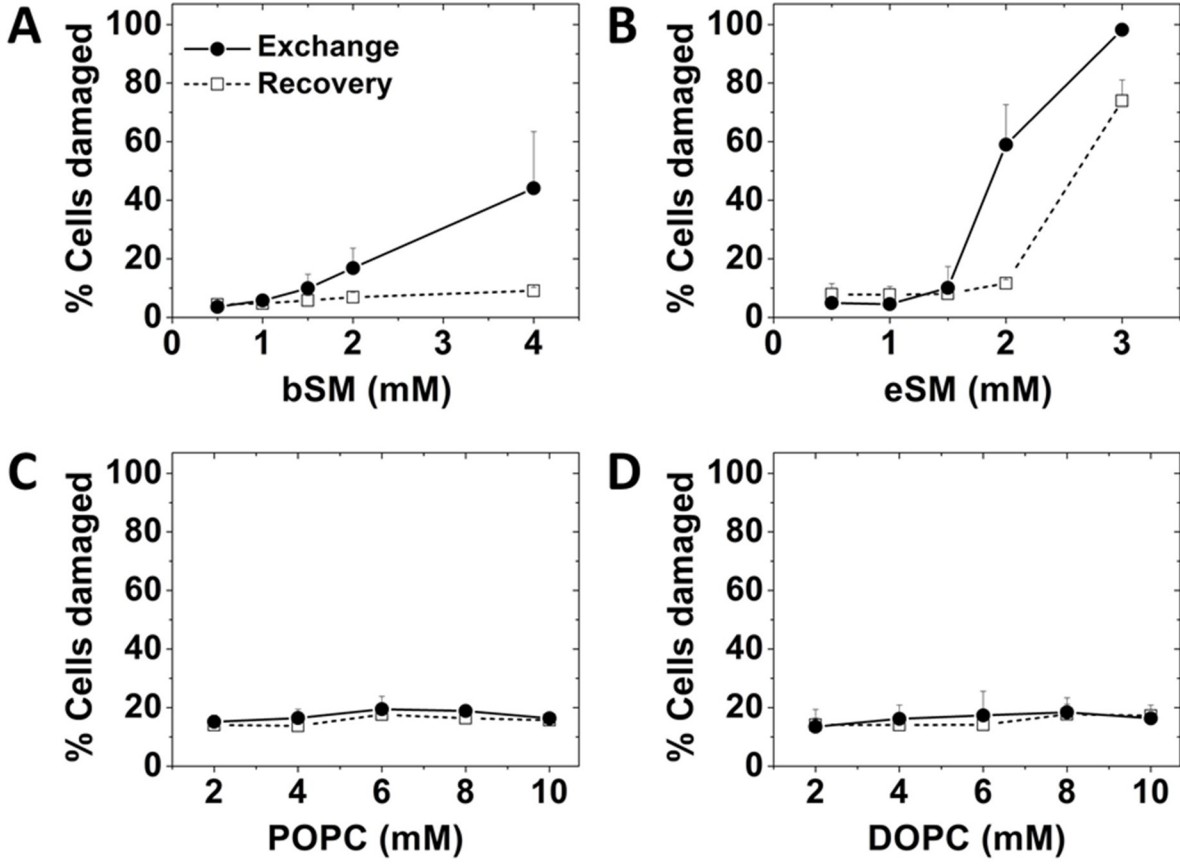

**Fig 8. Percent damaged A549 cells after 1h lipid exchange at 26˚C with or without a 2h recovery step at 37˚C assayed with PI and Annexin V.** Exchange was carried out using a range of concentrations of: A) bSM; B) eSM; C) POPC; D) DOPC. Exchange was carried out in 6 cm diameter plates at 26˚C with 1.5 ml of lipid plus 40mM MαCD. Recovery was carried out at 37˚C in the same plates with 2 ml of complete growth medium. In untreated samples there was about 5% damaged cells. Mean and standard deviation from three experiments is shown.

carried out at lower temperatures. Notice that incubation of cells with exogenous SM or PC did not alter SM levels noticeably in cells in controls carried out in the absence of MαCD.

Lipid localization after exchange was also assayed. To do this, radiolabeled lipids were exchanged into the outer leaflet, and after that a second round of exchange (i.e. back exchange) with unlabeled lipids was carried out. Fig 11 shows that a majority of lipid could be back exchanged out of cells, indicating localization in the plasma membrane (and compartments rapidly cycling to the plasma membrane) both immediately after exchange and recovery. This result is consistent with the changes in plasma membrane physical properties after exchange (see below). While the exogenous SM and POPC remained plasma membrane-localized after exchange, SM remained plasma membrane localized after recovery, while POPC showed less plasma membrane localization after 2h recovery. This may reflect transportation of POPC between various PC pools in the cell. Since SM is largely a plasma membrane lipid, it would not be expected to be greatly transported to internal membranes.

## Lipid exchange alters plasma membrane properties

Finally, the effect of lipid substitution on physical properties of cell membranes was measured (Table 1). Changes in plasma membrane properties after exchange and recovery were first

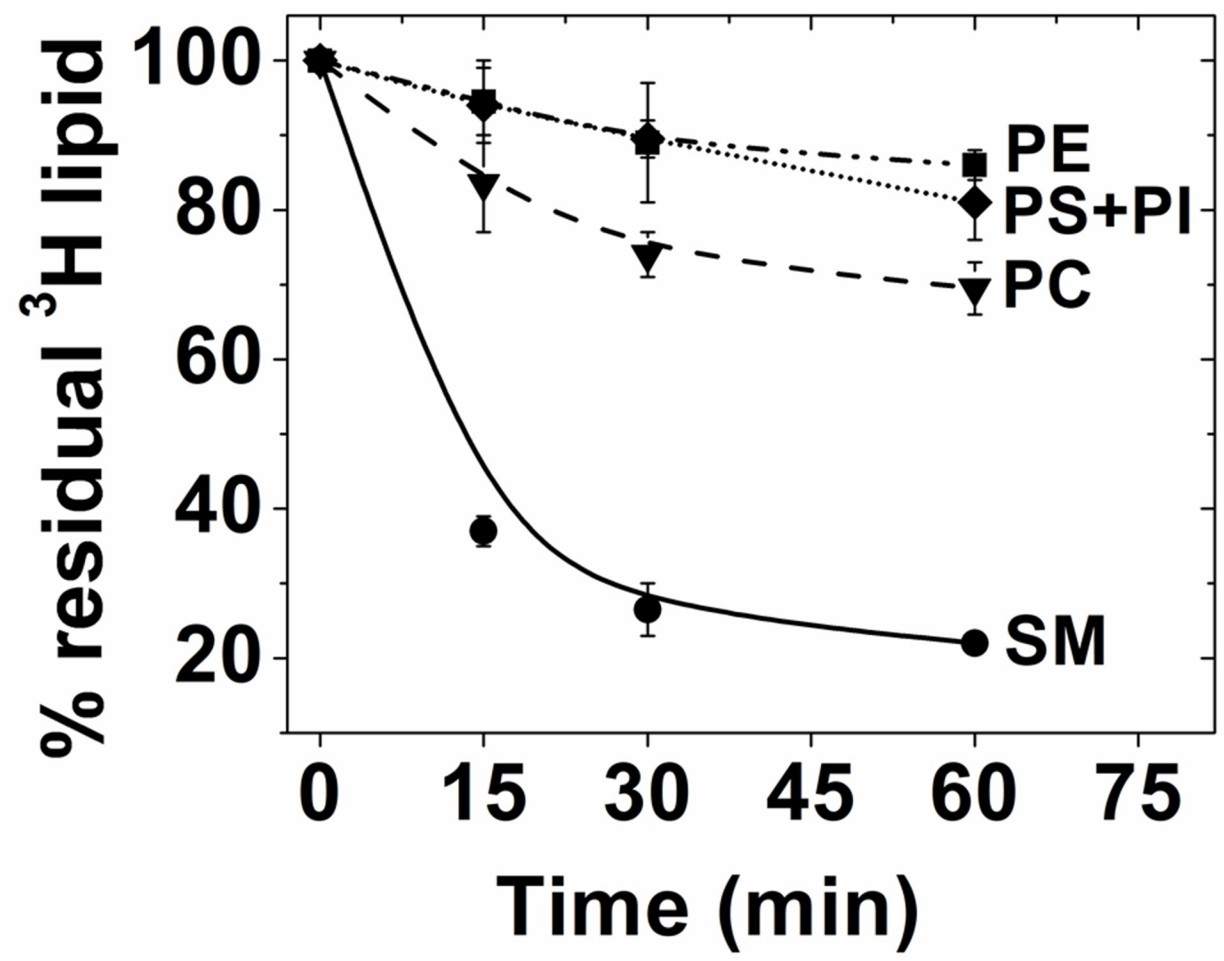

**Fig 9. Time-course of endogenous lipid exchange out of ³H labeled CHO cells.** Endogenous ³H lipids analyzed by measuring radioactivity in lipid TLC bands from a total lipid extract (for details see Materials and Methods). The residual % ³H lipid = [(cpm for a lipid after exchange)/(cpm for that lipid at time 0)] x 100% Exchange was carried out in 10 cm diameter plates at 37°C with 2 ml of 2 mM eSM plus 50 mM MαCD. Mean and range from two experiments is shown.

probed with Laurdan generalized polarization (GP), which probes membrane packing and hydration levels [17]. In the Lo state, GP values are higher than in the Ld state. Table 1 shows that after substitution with POPC or DOPC, A549 cells show lower GP values than after substitution with SM, in line with the expectation that substitution with unsaturated PC should decrease lipid packing. Membrane properties were also assessed using TMA-DPH, which associates with the plasma membrane outer leaflet. Its fluorescence anisotropy measures membrane packing [5], but not necessarily identically as Laurdan (see Discussion). Anisotropy is higher in the Lo state than in the Ld state. A decrease in anisotropy was measured upon substitution with POPC or DOPC relative to that after substitution with SM. Analogous experiments were carried out using giant plasma membrane vesicles (GPMV). GPMV are plasma membrane fragments that can be induced to pinch off from cells [21–24].

GPMV were prepared from RBL-2H3 cells [17], which yield high levels of GPMV. The effect of lipid substitutions on membrane packing and order in GPMVs was similar to that seen in the whole cells (Table 1). Combined, these experiments show that plasma membrane physical properties can be manipulated as expected by lipid substitution (see Discussion). As

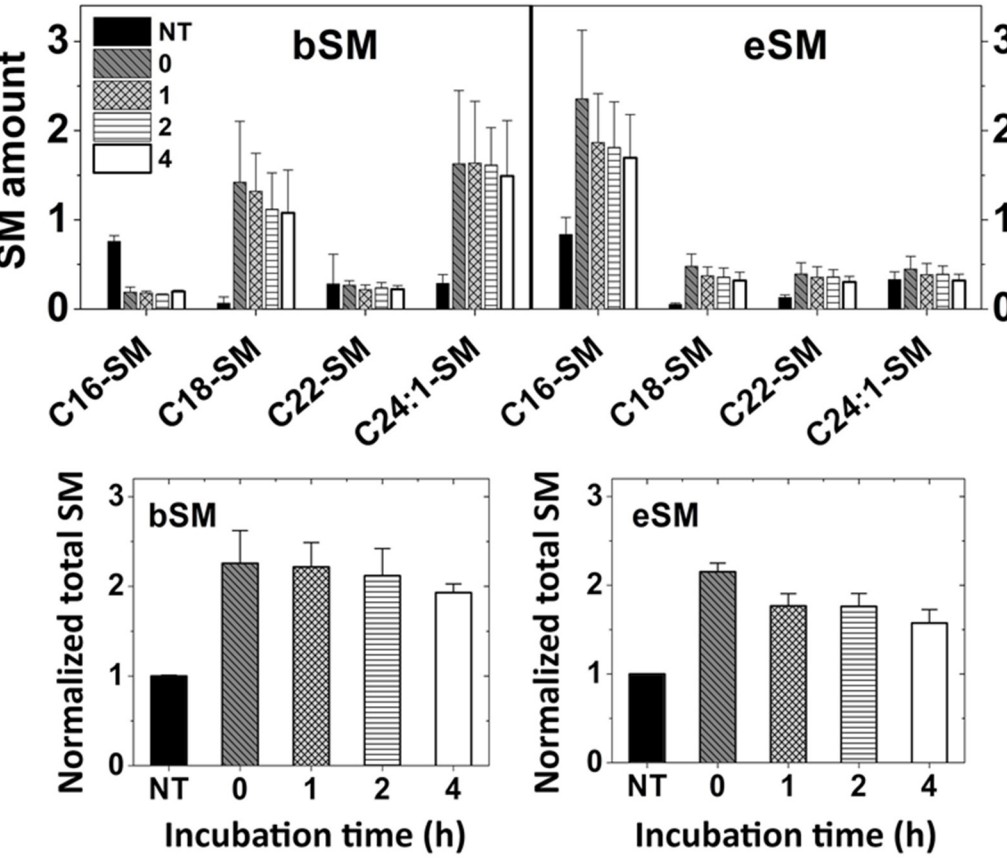

**Fig 10. SM levels remain elevated and SM acyl chains unaltered for hours after lipid exchange as assayed by mass spectrometry.** A549 cells were subjected for exchange using 1.5 mM eSM or bSM with 40 mM MαCD for 1 h at 37°C. The cells were harvested for lipid extraction and SM analyzed immediately aft MαCD er exchange or after 1, 2 or 4 h incubation in complete RPMI 1640 medium. A. Major SM species profiles after exchange with bSM (left panel) or eSM (right panel). (SM amount shown in (pmole x $10^4$)/mg protein)). B. Total SM levels after exchange with bSM (left panel) or eSM (right panel). SM levels shown relative to that in untreated cells. NT = untreated cell control. Exchange was carried out in 3.5 cm diameter plates at 37°C with 1 ml of lipid plus 40 mM MαCD. Post exchange incubation (recovery) was carried out at 37°C in the same plates with 1.5 ml of medium. Mean and standard deviation from three experiments is shown.

noted above, they also confirm that after lipid substitution and recovery plasma membrane lipid composition has been altered.

## Discussion

This report shows that lipid exchange conditions minimizing damage to mammalian cells can be achieved at optimal concentrations of lipid and MαCD, while concentrations that lead to an imbalance of lipid delivery and extraction damage cells. In RBC this can lead to lysis, and in cultured mammalian cells damage weakens cell membrane integrity, and as implied by changes in morphology, somehow alters in cytoskeletal-membrane interactions.

Importantly, we found that allowing cultured cells to recover in complete growth medium with serum after the exchange step, and/or choice of an appropriate exchange temperature can prevent/minimize cell damage. Although additional time is incurred by adding a recovery step, changes of plasma membrane composition (and physical properties) after lipid exchange can be largely maintained for several hours. Combined, these results expand the range of

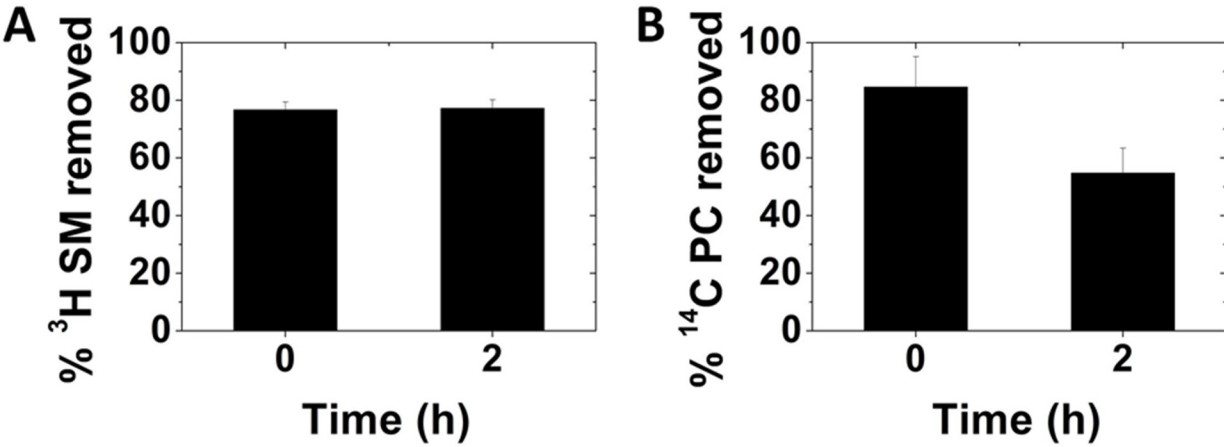

**Fig 11. Analysis of back exchangeability of radiolabeled SM or POPC introduced into A549 cells by lipid exchange.** (A) Back exchangeability of $^3$H-SM immediately after exchange (0) or 2 h recovery after exchange with 1:1 bSM:POPC. (B) Back exchangeability of $^{14}$C-DOPC immediately after exchange (0) or 2 h recovery after exchange with POPC. The p value for the significance of the difference at 0 and 2 h is 0.04. Back exchangeability is given by the % of radiolabeled lipid removed after a 1h second lipid exchange step in the presence of MαCD using unlabeled lipid. Exchange was carried out in 3.5 cm diameter plates with 1 ml of lipid plus 40 mM MαCD (See Methods for details). Recovery was carried out at 37˚C in the same plates with 1.5 ml of complete growth medium. Mean and standard deviations from three experiments are shown.

conditions under which efficient lipid exchange can be carried out upon cells without cell damage. As a result, this greatly extends the range of the conditions in which the effect of exchanging lipids can be usefully studied. Future studies to further expand the types of lipids (in terms of both headgroup and acyl chain structure) that can be used to replace the outer leaflet lipids without inducing cell damage should be of high priority.

It is noteworthy that in some cases lipid concentration was crucial while in other cases a very wide range of lipid concentrations could be used without affecting the extent of cell damage. We speculate that in the latter cases the balance between the amount of lipid delivered and extracted is not very dependent upon the lipid to MαCD ratio. A possible explanation for this is that in some cases the binding of lipid to MαCD vs. lipid binding to a membrane may be very sensitive to whether there is a lipid excess or lipid deficiency in the outer leaflet. A lipid

**Table 1. Effect of lipid exchange with SM or PC on membrane physical properties.**

|  | No exchange | bSM | POPC | DOPC |
|---|---|---|---|---|
| A 549 cells | | | | |
| Laurdan GP | 0.182 ± 0.001 | 0.188 ± 0.009 | 0.175 ± 0.006 | 0.168 ± 0.007 |
| TMADPH anisotropy | 0.324 ± 0.010 | 0.327 ± 0.012 | 0.305 ± 0.013 | 0.308 ± 0.004 |
| RBL-2H3 GPMV | | | | |
| Laurdan GP | 0.389 ± 0.024 | 0.445 ± 0.015 | 0.256 ± 0.036 | 0.264 ± 0.013 |
| TMADPH Anisotropy | 0.276 ± 0.002 | 0.277 ± 0.004 | 0.261 ± 0.008 | 0.262 ± 0.008 |

Exchange was carried out in A549 cells using 3.5 cm diameter plates at ~25˚C with 1ml of 1.5 mM bSM, or 3mM PC and 40 mM MαCD or in RBL-2H3 cells using 10 cm diameter plates at ~25˚C and 2 ml of the same lipid concentrations plus 40 mM MαCD. Recovery was carried out at 37˚C in the same plates with 1.5 ml of complete growth medium for the 3.5 cm diameter plates or 10 ml for the 10 cm diameter plates. Mean and standard deviation for three experiments is shown. In cells, for the difference in GP between SM and POPC p = 0.057 and for SM and DOPC p = 0.021. For the difference in anisotropy between SM and POPC p = 0.027 and between SM and DOPC p = 0.037. In GPMV, for the difference in GP between SM and POPC p = 0.003 and between SM and DOPC p<0.001. For the difference in anisotropy between SM and POPC p = 0.026 and for SM and DOPC p = 0.033. For the difference in GP between no exchange and SM p = 0.016

excess in the outer leaflet would cause it to be crowded, and thus decrease its ability to bind lipid from the MαCD, while a lipid deficiency would leave a gap in the outer leaflet, and would increase the tendency of the outer leaflet to bind lipid. If these effects are large with even small amounts of imbalance, it would tend to keep the inner and outer leaflet in near balance during exchange.

In terms of changes in cell membrane properties, the decrease in membrane order upon replacement of endogenous SM with an unsaturated PC seen after lipid exchange is as expected from the behavior of these lipids in lipid vesicles [5], i.e. introduction of SM results in higher membrane order than introduction of PC. It was somewhat of a surprise that there was no (statistically significant) increase in order relative to untreated cells when SM was exchanged into cell membranes. One possibility is that the physical state of the bilayer is close to that of the Lo state even before exchange. However, the low sensitivity of assays of membrane order may be an issue. Preliminary FRET studies on GPMV suggest that ordered domain formation relative to untreated controls is significantly thermally stabilized by increasing SM content, as well as thermally destabilized by increasing unsaturated PC content (Q. Wang and E. London, unpublished observations).

It was also noteworthy that Laurdan and TMA-DPH detected similar but not identical changes in membrane order. This may reflect some difference in what the two probes measure. When added to membranes, TMA-DPH, which is anchored at one end to the membrane surface by its charge, is embedded in the outer membrane leaflet [5, 25] and so measures plasma membrane outer leaflet membrane order. In contrast, Laurdan is likely to be localized in both leaflets of the plasma membrane, as well as in internal membranes. This would explain why it detected lower membrane order in whole cells than in GPMV. In the case of TMA-DPH, the partial loss of asymmetry in GPMV [26], which would increase the outer leaflet content of unsaturated lipids, may explain the slightly lower level of order it detects in GPMV relative to that it detects in whole cells. In addition Laurdan fluorescence is sensitive to both differences in packing and changes in hydration, which can indirectly reflect changes in packing [27].

## Supporting information

**S1 Fig. Difference in Cell Morphology for PI/Annexin V positive and PI/Annexin V negative A549 cells after lipid exchange at 26˚C.** Green: PI and Annexin V negative cells. Red: PI and Annexin V positive cells. Exchange was carried out in 6 cm diameter plates at 26˚C with 1.5 ml of with 2 mM bSM and 40 mM MαCD. Numbers at top show % of double negative and double positive cells out of total cells. Total number of cells counted was 50,000, including singly positive cells.
(TIF)

**S2 Fig. Cell viability assay after lipid exchange and recovery. Bars illustrated percent of trypan blue-negative CHO cells after exchange using SM at concentrations shown.** Exchange was carried out in 3.5 cm diameter plates at 37˚C with 1 ml of lipid plus 50 mM MαCD. Mean and standard deviation from three experiments is shown.
(TIF)

**S3 Fig. Rate of endogenous [3]H SM exchange out of radiolabeled cells is similar at 27˚C and 37 ˚C.** A549 endogenous lipids were labelled with [3]H and lipid exchange carried out with 1 mM exogenous bSM and 40 mM MαCD. Residual endogenous [3]H labelled SM was monitored by lipid extraction from cells every 15 min after exchange initiated. Time 0 was [3]H labelled A549 cells incubated with serum-free growth medium for 1 h. Exchange was carried out in 10 cm diameter plates at temperature shown with 3 ml of lipid plus 40 mM MαCD. Mean and

standard deviation from three experiments is shown.
(TIF)

**S4 Fig. Example of TLC analysis of lipids after lipid exchange at 27˚C.** HP-TLC of A549
cells after 1h exchange and 2h recovery. Cells were incubated with 1 mM SM or 4 mM PC
(exogenous lipid type shown under TLC) without MαCD or in exchange medium containing
lipid vesicles mixed with MαCD. Levels of SM quantified using imageJ densitometry scan and
are shown above the SM bands (see arrow). Exchange was carried out in 10 cm diameter plates
at 27˚C with 3 ml of lipid plus 40 mM MαCD. Recovery was carried out at 37˚C in the same
plates with 5 ml of complete growth medium. Similar results were observed in a second TLC
experiment.
(TIF)

**S5 Fig.** Raw flow cytometry data with axis values shown for: A. Fig 3C or B. Fig 5B.
(TIF)

## Author Contributions

**Conceptualization:** Erwin London.

**Funding acquisition:** Erwin London.

**Investigation:** Guangtao Li, Shinako Kakuda, Pavana Suresh, Daniel Canals, Silvia Salamone.

**Methodology:** Guangtao Li, Shinako Kakuda, Pavana Suresh, Daniel Canals, Silvia Salamone.

**Project administration:** Erwin London.

**Supervision:** Erwin London.

**Writing – original draft:** Guangtao Li, Shinako Kakuda, Pavana Suresh, Daniel Canals, Erwin
London.

**Writing – review & editing:** Guangtao Li, Shinako Kakuda, Pavana Suresh, Daniel Canals,
Erwin London.

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
