## [Decision Letter · Decision Letter 0]

25 Sep 2019

Replacing the Plasma Membrane Outer Leaflet Lipids With Exogenous Lipid Without Damaging Membrane Integrity

PONE-D-19-18785

Dear Dr. London,

I apologize for the tardy decision concerning your manuscript, we were waiting for an additional review that never came. However, based on the review we did receive, we are pleased to inform you that your manuscript has been judged scientifically suitable for publication and will be formally accepted for publication once it complies with all outstanding technical requirements. The manuscript was well written but nonetheless I encourage you to proofread the manuscript again for thoroughness sake.

With kind regards,

Christopher Beh, PhD

Academic Editor

PLOS ONE

Additional Editor Comments (optional):

Reviewers' comments:

Reviewer's Responses to Questions

**Comments to the Author**

1. Is the manuscript technically sound, and do the data support the conclusions?

Reviewer #1: Yes

2. Has the statistical analysis been performed appropriately and rigorously? 

Reviewer #1: Yes

3. Have the authors made all data underlying the findings in their manuscript fully available?

Reviewer #1: Yes

4. Is the manuscript presented in an intelligible fashion and written in standard English?

Reviewer #1: Yes

5. Review Comments to the Author

Reviewer #1: Li et al provide a nice and very important extension of their previously reported lipid exchange technique for the outer leaflet of plasma membranes. In particular, they detail on proper experimental conditions to avoid or even recover cell damage induced during the exchange. This is outstanding work. I have no further ideas for any improvement of the manuscript. I recommend publication as is.

6. PLOS authors have the option to publish the peer review history of their article (what does this mean?). If published, this will include your full peer review and any attached files.

Reviewer #1: No

---

## [Editor Report · Acceptance letter]

30 Sep 2019

PONE-D-19-18785 

Replacing the Plasma Membrane Outer Leaflet Lipids With Exogenous Lipid Without Damaging Membrane Integrity 

Dear Dr. London:

I am pleased to inform you that your manuscript has been deemed suitable for publication in PLOS ONE. Congratulations! Your manuscript is now with our production department. 

With kind regards,

on behalf of

Dr. Christopher Beh 

Academic Editor

PLOS ONE